# Mental health laws in Africa: Perspectives from Cabo Verde, Egypt, Ghana, Kenya and Nigeria

Deborah Oyine Aluh[1,2,3] 🔘, Wisdom Joe Igbokwe[4], Emmanuel Nii-Boye Quarshie[5,6], Ana Moniz[7], Berrick Otieno[8] and AbdulRahman A. Saied[9,10]

[1]Department of Clinical Pharmacy and Pharmacy Management, University of Nigeria Nsukka, Nsukka, Enugu State, Nigeria; [2]Lisbon Institute of Global Mental Health, Comprehensive Health Research Centre (CHRC), Nova Medical School, Nova University of Lisbon, Portugal; [3]School of Health, Science and Society, University of Greater Manchester, United Kingdom; [4]Clinical Sciences Department, Nigerian Institute of Medical Research; [5]Department of Psychology, University of Ghana, Accra, Ghana; [6]Department of Psychology, University of Johannesburg, Johannesburg, South Africa; [7]UCNVA da Unipiaget, de Cabo Verde; [8]Institute for Human Development, Aga Khan University, Nairobi, Kenya; [9]Ministry of Tourism and Antiquities, Aswan Office, Aswan 81511, Egypt and [10]Aswan Research Group, Aswan, Egypt

## Research Article

**Keywords:**
mental health legislation; UNCRPD; Africa; context; human rights

**Corresponding author:**
Deborah Oyine Aluh;
Email: aluhdeborah@yahoo.com

## Abstract

Mental health legislation across Africa has evolved significantly from colonial-era frameworks. An adapted version of the FOSTREN* (Fostering and Strengthening Approaches to Reducing Coercion in European Mental Health Services) instrument, which is a comprehensive assessment tool based on the World Health Organisation Mental Health Legislation Checklist and the United Nations Convention on the Rights of Persons with Disabilities, was used to analyse mental health laws from Nigeria, Egypt, Ghana, Cabo Verde and Kenya. The comparative analysis showed varying alignment with international human rights standards, reflecting complex interactions between global frameworks and local realities. All the mental health laws analysed show movement towards rights-based approaches, although implementation challenges related to resource constraints, service delivery capacity and cultural integration remain significant barriers. Ghana's formal integration of complementary and alternative medicine into its mental health framework, which requires cooperation between the Mental Health Authority and Traditional and Alternative Medicine Council, and the inclusion of people with lived experience of mental health conditions in review panels are examples of innovative approaches that show promise for regional adoption. While some form of supported decision-making is available, none of the countries offer advanced care directives. The study highlights that legislative reform alone is insufficient without addressing contextual factors like poverty, healthcare financing and integration of traditional healing practices in developing rights-based mental health care systems.

## Resumo

A legislação sobre saúde mental em toda a África evoluiu significativamente desde os quadros da era colonial. Uma versão adaptada do instrumento FOSTREN*, que é uma ferramenta de avaliação abrangente baseada na Lista de Verificação da Legislação sobre Saúde Mental da OMS e na UNCRPD, foi utilizada para analisar as leis de saúde mental da Nigéria, Egito, Gana, Cabo Verde e Quénia. A análise comparativa mostrou um alinhamento variável com as normas internacionais de direitos humanos, refletindo interações complexas entre os quadros globais e as realidades locais. Todas as leis de saúde mental analisadas mostram um movimento em direção a abordagens baseadas em direitos, embora os desafios de implementação relacionados com restrições de recursos, capacidade de prestação de serviços e integração cultural continuem a ser barreiras significativas. A integração formal do Gana da medicina complementar e alternativa no seu quadro de saúde mental, que requer a cooperação entre a Autoridade de Saúde Mental e o Conselho de Medicina Tradicional e Alternativa, e a inclusão de pessoas com experiência vivida de condições de saúde mental em painéis de revisão são exemplos de abordagens inovadoras que se mostram promissoras para a adoção regional. Embora exista alguma forma de apoio à tomada de decisões, nenhum dos países oferece diretivas de cuidados avançados. O estudo destaca que a reforma legislativa por si só é insuficiente sem abordar fatores contextuais como a pobreza, o financiamento dos cuidados de saúde e a integração das práticas de cura tradicionais no desenvolvimento de sistemas de cuidados de saúde mental baseados nos direitos.

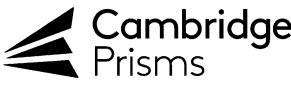


## Impact statement

Mental health legislation across Africa stands at a critical juncture where progressive legal frameworks diverge from implementation realities. This study examines mental health laws in

five African countries, highlighting the tensions in balancing international human rights standards with local cultural realities. The findings inform mental health reform efforts across the continent by identifying successful innovations that can be replicated. Ghana's inclusion of people with lived experience of mental health conditions in review panels demonstrates how African countries can lead global best practices in participatory governance. Egypt's comprehensive monitoring systems show that effective oversight is achievable even with limited resources. The tension between individual autonomy and communitarian decision-making traditions highlights the need for more culturally responsive approaches to mental health law development. The exclusion of traditional and faith-based mental health care from formal legal frameworks despite their widespread use is a critical gap that affects millions of Africans seeking mental health care. For international donors and development organisations, the study demonstrates that legislative reform must be accompanied by investments in mental healthcare infrastructure, professional training and poverty reduction to be effective. The findings contribute to ongoing debates about decolonising mental health systems and developing culturally responsive, rights-based services that honour both universal human rights principles and indigenous knowledge.

## Introduction

Approximately 15% of the global population, totalling over one billion individuals, live with disabilities and face inequalities, exclusion and discrimination. Among these, people with mental health and psychosocial disabilities encounter particularly severe social challenges and remain vulnerable to human rights abuses worldwide (Hoffman et al., 2016). Mental health laws and policies are typically developed to safeguard the rights of persons with mental health and psychosocial disabilities. The United Nations Convention on the Rights of Persons with Disabilities (UNCRPD), adopted in 2008, is the most widely ratified legal framework protecting the rights of people with disabilities. It advocates fundamental principles, including respect for inherent dignity, individual autonomy and freedom to make personal decisions, non-discrimination, full participation in society, equal opportunity and accessibility (Drew et al., 2016; Steinert et al., 2016). The Committee on the Rights of Persons with Disabilities, established under Article 34 of the UNCRPD to monitor implementation, has issued authoritative interpretations through General Comments expected to significantly shape mental health law reform globally. General Comment No. 1 on Article 12 (equal recognition before the law) states that involuntary detention and forced treatment violate the right to legal capacity and calls for abolition of substitute decision-making regimes in favour of supported decision-making (CRPD/C/GC/1, 19 May 2014) (United Nations Committee on the Rights of Persons with Disabilities General Comment No 1: Article 12: Equal Recognition before the Law, 2014). The Committee also established that mental health services must be community-based and provided with free and informed consent (General Comment No. 5 on Article 19), and that any deprivation of liberty based on disability or mental health grounds constitutes discrimination and violates the Convention (Guidelines on Article 14) (Committee on the Rights of Persons with Disabilities Pronounces on Article 14 – Mental Capacity Law and Policy, 2014; General Comment No. 5 on Article 19 – the Right to Live Independently and Be Included in the Community | OHCHR, 2017). These interpretations have sparked a movement to reform mental health laws, influencing signatory countries to enact legislation that prioritises human rights, reduces coercion and promotes supported decision-making (McCusker *et al.*, 2023). However, they have also generated debate about practical implementation, particularly in resource-limited settings and diverse cultural contexts.

In Africa, the landscape of mental health legislation is diverse and evolving. The African Charter on Human and Peoples' Rights (ACHPR), one of the continent's foundational human rights instruments, came into force on October 21, 1986. Article 16 of the ACHPR establishes the right to health, obligating member states to protect both physical and mental well-being. While the Charter does not provide detailed guidance on mental health, it provides a foundation for advocating rights-based mental health care across the continent (African Charter on Human and Peoples' Rights | African Union, 1981). Similarly, the Protocol to the African Charter on the Human and People's rights on the Rights of Persons with Disabilities, which was adopted in 2018 and came into force in May 2024, aligns with the UNCRPD, in calling for progressive laws to address discrimination and promote inclusion for people with disabilities, including those with mental health conditions (Dey et al., 2019). Colonial-era mental health laws, which emphasised institutionalisation as the primary response to mental health conditions, continue to influence many African countries. These historical systems often resulted in coercive practices and continue to shape contemporary approaches to mental health care in many jurisdictions (Di Pierdomenico et al., 2024; Juma and Ngwena, 2024). However, global frameworks like the UNCRPD are now driving a shift towards rights-based mental health legislation. All African countries have now ratified the UNCRPD, prompting significant legislative reforms in some countries.

Understanding how African countries are implementing these international frameworks within their specific contexts is essential for evaluating progress towards rights-based mental health care. This analysis is particularly important given the potential tensions between universalist human rights frameworks and diverse African cultural approaches to mental health care, family involvement and decision-making processes. The widespread reliance on traditional medicine and spiritual healing across Africa, particularly in rural areas where these practices often serve as the first line of mental health care (Assad et al., 2015; Anjorin and Hassan Wada, 2022; Okasha et al., 2024), makes their integration into formal systems a critical consideration for rights-based legislation. The emphasis on individual rights, central to international frameworks like the UNCRPD, creates challenges in communitarian African societies where collective decision-making processes and family involvement in healthcare decisions are traditionally valued (Ewuoso, 2018; Aluh et al., 2022). Understanding how different countries navigate these tensions while working towards human rights compliance provides important insights for mental health law development across the continent.

Previous studies have compared mental health legislation across jurisdictions, exploring various themes. One study comparing mental health legislation from diverse commonwealth jurisdictions assigned an autonomy score rating, a numerical measure that evaluates the level of autonomy granted by mental health legislation. The study highlighted the ongoing tension between balancing patients' rights and providing mental health care (Fistein et al., 2009). Another study comparing mental health laws in five high-income countries found that they were influenced by, and moving towards alignment with, international standards like the UNCRPD. Some differences were observed among jurisdictions in criteria for

involuntary treatment, review timelines and availability of supported decision-making (Cronin et al., 2017). Rains and colleagues explored the association between involuntary hospitalisation rates and legal framework in 22 European countries and Australia and New Zealand. They observed significant variations in annual involuntary hospitalisation rates, although no clear association between legislative frameworks and involuntary hospitalisation rates was found. Higher annual rates of hospitalisation were weakly associated with lower poverty rates, higher Gross Domestic Product per capita and a higher number of inpatient beds (Sheridan Rains et al., 2019). In 2013, the mental health laws in Ghana, South Africa, Uganda and Zambia were assessed using the WHO Checklist on Mental Health Legislation (Drew et al., 2013). The evaluation revealed that older laws in Ghana, Uganda and Zambia failed to address the fundamental human rights of people with mental health conditions. While South Africa's Mental Health Care Act (2002) incorporated critical human rights standards, certain provisions inadequately safeguarded against potential violations (Drew et al., 2013). In South Asia, a comparison of the mental health legislation in India, Pakistan, Bangladesh and Sri Lanka revealed that all the countries except Sri Lanka had reformed their laws to align more with international standards. However, contextual factors, such as resource limitations and cultural norms, were noted to hinder full implementation (Dey et al., 2019), as has been noted elsewhere (Aluh et al., 2023; Whittington et al., 2023).

While previous studies have provided valuable insights on comparing mental health legislation globally, there is a lack of recent comparative research focusing on African countries. The interplay between mental health laws and broader contextual factors, such as sociopolitical, economic and cultural factors, are often underexplored. This study aims to fill this gap by comparing the mental health laws of five African countries (Cabo-Verde, Egypt, Ghana, Kenya and Nigeria) and to contextualise the findings in relation to universal health coverage, poverty headcount, year of independence and democratic governance. The research questions guiding this study are: How do the mental health laws of the selected African countries align with international human rights standards?; What are the key similarities and differences in the mental health legislation of these countries?; and What are the contextual factors within which these countries seek to legislate in alignment with international frameworks their governments have signed up to?

## Methods

### Study design and country selection

This study employed a questionnaire-based survey to gather information about the mental health laws of five purposively selected African countries, representing different regions on the continent. The countries were selected to ensure geographical, linguistic and cultural representation across Africa. Ghana, Kenya and Nigeria were selected as English-speaking countries with shared British colonial histories. Egypt was included due to its significant historical foreign influence, particularly from the British, and its use of Modern Standard Arabic in formal contexts. Cabo Verde, a Lusophone country with Portuguese colonial heritage, was selected to represent a different linguistic and cultural context. This selection strategy enabled analysis across diverse legal traditions and colonial influences that continue to shape contemporary mental health legislation. Among the study countries, Egypt and Kenya were early adopters of the UNCRPD, ratifying it in 2008, followed by Nigeria (2010), Cabo Verde (2011) and Ghana (2012). However,

compliance with reporting obligations has been inconsistent. Only Ghana and Kenya have received Concluding Observations from the CRPD Committee following the timely submission of their initial reports. Progress on the regional African Disability Protocol is also limited, with only Kenya and Nigeria having ratified this instrument.

### Countries' context

Understanding the contextual factors is essential for interpreting the mental health laws across these five countries. As shown in Table 1, the countries vary significantly in their socioeconomic indicators and healthcare infrastructure. Nigeria allocates 4.1% of its total health budget to mental health, while Ghana directs ~1.4% of its total health expenditure to mental health services. Cabo Verde and Egypt provide the most comprehensive insurance coverage for mental health services. Kenya has the highest poverty rate, with 36.1% of the population living on less than $2.15 per day. The Human Rights Index, which captures freedom from government torture, political killings, forced labour, property rights and freedoms of movement, religion, expression and association, ranks Egypt lowest (0.27) and Cabo Verde highest (0.92).

### Questionnaire adaptation and validation

An abridged version of the FOSTREN (Fostering and Strengthening Approaches to Reducing Coercion in European Mental Health Services) instrument for evaluating mental health laws and policies was adapted for this study (Aluh et al., 2024). The FOSTREN instrument was originally developed by a European network dedicated to reducing coercion in mental health care. Given that this instrument was developed primarily within European contexts, careful adaptation was necessary to ensure its relevance to African legal and cultural contexts. Questions on Community Treatment Orders, which were unavailable in the selected countries, were removed, while new items were added to address complementary and alternative medicine (CAM) and traditional healing practices, given their significant role in mental health care across Africa. Contextual indicators such as Universal Health Coverage status and poverty levels were also incorporated. The adapted instrument included variables measuring legal criteria for involuntary admission, authorisation processes, patient rights and protections, regulation of coercive measures (seclusion, restraints and treatments like electroconvulsive therapy [ECT]), oversight mechanisms and integration with traditional and alternative medicine practices. The questionnaire was first piloted in Ghana and Nigeria to identify any ambiguities or difficulties in understanding the questions.

### Data collection

Data were collected between March 2024 and August 2024. Respondents for each country were identified through the lead author's professional network and included mental health nurses and researchers from Kenya and Cabo Verde, a clinical psychologist and researcher from Ghana and public health researchers from Nigeria and Egypt. All authors share a keen interest in mental health laws and policies. Respondents were encouraged to collaborate with local stakeholders to address any informational gaps. The respondents were required to respond to the survey as provided by their respective mental health laws, focusing on the exact wording of the law rather than their subjective interpretations. Data about contextual factors were obtained from resources including

**Table 1.** Countries' context

| Variables | Nigeria | Ghana | Kenya | Cabo Verde | Egypt |
|---|---|---|---|---|---|
| Year of independence | 1960 | 1957 | 1963 | 1975 | 1922 |
| Population | 227,882,945 | 33,787,914 | 55,339,003 | 522,331 | 114,535,772 |
| Life expectancy[a] | 63.4 years | 66.1 years | 66.8 years | 73.2 years | 69.1 years |
| Number of psychiatrists | <300 | 39 | 115 | 7 | 1,106 |
| Budget for mental health services | 4.1% | 1.4% | Not available | Not available | <2% |
| Payment for mental health care | People pay mostly or entirely out of pocket for services | People pay at least 20% towards the cost of services | People pay mostly or entirely out of pocket for services | People pay nothing at the point of service use (fully insured) and medicines | Not all services are fully covered under health insurance |
| Standalone hospitals | 8 (Federal), 15 (State) | 3 | 1 | Not available | 22 |
| Mental health services in general hospitals | Not available | 260 | 14 | 3 | 13-18 mental health departments within general and central hospitals |
| Community MHS | Not available | 1,026 | Not available | 1 | 62-80 |
| Poverty headcount ratio[b] | 30.9 | 25.2 | 36.1 | 4.6 | 27.3 |
| Human Rights Index 2024[c] | 0.68 | 0.9 | 0.72 | 0.92 | 0.27 |

*Note:* Poverty headcount ratio at $2.15 a day (2017 PPP) (% of population).
[a]https://data.who.int/countries.
[b]https://data.worldbank.org/indicator/SI.POV.NAHC?end=2021&start=2021&type=shaded&view=map.
[c]https://ourworldindata.org/human-rights.

the World Health Organisation (WHO) Mental Health ATLAS and World Bank dashboards. Following questionnaire completion, responses were reviewed by the lead and second authors, who cross-referenced them against published mental health legislation from each country. This process involved a detailed examination of legal documents to verify the accuracy and completeness of reported information. Respondents from each country were then invited to review the compiled results and provide clarifications on any discrepancies or concerns that arose during the compilation and analysis phase.

## Results

Each of the five countries initially adopted mental health laws inherited from the colonial era, which historically emphasised custodial care and institutional management of people with mental health conditions, often with limited attention to human rights and well-being. However, all countries in our survey have enacted new legislation or updated some aspects of their mental health laws within the past 15 years, reflecting growing international recognition of human rights principles in mental health care.

### *Regulation of involuntary admission*

All five countries have dedicated mental health Acts that regulate voluntary and involuntary admissions of people with mental health and psychosocial conditions. As detailed in Table 2, the most recent is the Nigerian National Mental Health Act of 2021 (Saied, 2023).

All countries use risk of harm to self or others as a primary legal criterion for involuntary admission. However, only Ghana explicitly considers the patient's decision-making capacity as an additional factor in determining the appropriateness of involuntary admission. The authorisation process shows similarities across countries, typically requiring assessment by qualified mental health practitioners and independent oversight bodies. Nigeria and Cabo Verde explicitly state that involuntary admission should be used as a last resort, while this principle is only implied in Ghana and Kenya's legislation and is not mentioned in Egypt's Act. The principle of least restrictive alternative is explicitly stated in all countries except Kenya and Egypt. Individual treatment planning is mandatory in all acts, except Kenya's. Most countries provide patients with the right to appeal involuntary admission to independent review bodies, such as mental health tribunals or specialised courts. Nigeria, Cabo Verde, Egypt and Ghana require periodic reviews of involuntary admissions, whereas Kenya does not mandate such reviews.

### *Protection and promotion of the rights of people with mental health and psychosocial conditions*

Table 3 presents the rights protections afforded to people with mental health conditions across the five countries.

Involuntarily admitted patients have the right to free legal representation in all countries, except Ghana. All countries require that patients be informed of the reasons for their involuntary admission, and their right to appeal; however, the Kenyan law does not require that patients be informed of their right to appeal.

**Table 2.** Regulation of involuntary admission in the mental health laws of five African countries

| Variables | Nigeria | Ghana | Kenya | Cabo Verde | Egypt |
|---|---|---|---|---|---|
| Mental health legislation | National Mental Health Act, 2021 | Act No. 846 of 2012, Mental Health Act | The Mental Health (Amendment) Act, 2022 | Lei n° 37/VIII/2013 | Law No. 71/2009 - Mentally Ill Patient Care (Mental Health Act) Decree 128/2010 - MOH – Executive regulations. Law No. 210/2020 … Amending provisions of the mentally ill patient care law No. 71/2009 Decree No. 55/2021 … |
| Year | 2023 | 2012 | 2022 | 2013 | 2009 |
| Legal criteria for involuntary admission | - Evidence of mental disorder. - Evidence that the mental health condition is so severe that failure to admit the person is likely to hinder the provision of appropriate treatment that can only be given by admission to a facility in accordance with the principle of the least restrictive alternative. - Serious risk of harm to self or others -Substantial risk of serious deterioration in the patient's condition if treatment is not given | - Evidence of mental disorder - Serious risk of harm to self or others - Substantial risk of serious deterioration in the patient's condition if treatment is not given - Prospect for substantial improvement in the patient's condition if treatment is given - The patient lacks decision-making capacity | - Evidence of mental disorder - Serious risk of harm to self or others - Substantial risk of serious deterioration in the patient's condition if treatment is not given | - Evidence of mental disorder - Serious risk of harm to self or others - It is the only way to ensure the interned person's treatment. | - Evidence of mental disorder of specified severity, such as psychosis or symptoms with equal impact (if yes, how is it specified) - Serious risk of harm to self or others - Substantial risk of serious deterioration in the patient's condition if treatment is not given? |
| Authorisation authority | - The responsible clinician must be a psychiatrist - An independent authority (e.g., review body or tribunal) | - The responsible clinician must be a psychiatrist - An independent authority | - The responsible clinician who can be a psychiatrist, medical practitioner, psychologist, clinical officer, psychiatric nurse | - The responsible clinician should be a psychiatrist - An independent authority - Legal representative of the person with a mental disorder | - The responsible clinician must be a psychiatrist - The responsible clinician who can be a general practitioner - An independent authority (e.g., review body* or tribunal) |
| Individual treatment plan required | Yes | Yes | No | Yes | Yes |
| involuntary admission specified as last resort | Yes (explicit) | No (implied) | No (implied) | Yes (explicit) | No (Implied) |
| Least restrictive principle | Yes | Yes | Yes | Yes | Yes |
| Right to appeal involuntaryadmission | Yes | Yes | No | Yes | Yes |
| Appeal recipients | An independent review body | An independent review body | Not applicable | Court of appeal | - An independent review body - Court |
| Time-bound periodic reviews of involuntary admission | Yes | Yes | No | Yes | Yes |

*Note:* MOH: Ministry of Health. Kenya's Mental Health (Amendment) Act, 2022, represents minimal amendments to the Mental Health Act of 1989, which was based on the English Mental Health Act of 1959.

Family notification requirements vary significantly. Nigeria's Mental Health Act mandates informing a patient's legal representative, either a formally appointed advocate or, in order of priority, the patient's spouse, adult children, parents or a court appointee, about the reasons for admission and their appeal rights. Ghana, Cabo Verde and Egypt mandate informing families of both

**Table 3.** Protection and promotion of the rights of people with mental health and psychosocial conditions in the mental health laws of five African countries

| Rights provisions | Nigeria | Ghana | Kenya | Cabo Verde | Egypt |
|---|---|---|---|---|---|
| Free legal representation | Yes | No | Yes | Yes | Yes |
| Informing patients of reasons for admission | Yes | Yes | Yes | Yes | Yes |
| Informing patients of the right to appeal | Yes | Yes | No | Yes | Yes |
| Informing family of the reasons for admission | Yes | Yes | Yes | Yes | Yes |
| Informing family of their rights to appeal | No | Yes | No | Yes | Yes |
| Informing legal representatives of the reasons for admission | Yes | Yes | Yes | Yes | No |
| Informing legal representatives of their rights to appeal | Yes | Yes | No | Yes | No |
| Non-legal advocacy services | No | No | No | No | Yes |
| Research participation protections | Not specified | Not specified | Not specified | Explicitly allows for choosing whether or not to participate in all forms of research | Explicitly prohibits participation |
| Advanced care directives | No | No | No | No | No |
| Supported decision-making | Yes | Yes | Yes | No | Yes |
| Judicial or quasi-judicial body to review processes | Yes | Yes | No | Yes | yes |
| Composition of the judicial or quasi-judicial body | - Legal practitioner with at least 10 years post-call experience<br>- Health care practitioner<br>- Lay person | - Legal practitioner<br>- Health care practitioner<br>- Service users or people with lived experience | NA | - Legal practitioner<br>- Health care practitioners<br>- Lay people | - Legal practitioner<br>- Health care practitioner<br>- Layperson reflecting the "community" perspective and others like nursing, human rights advocates |
| Allow appeal of the decision by this body to a higher court | Yes | Yes | NA | Yes | Yes |

*Note:* NA: not applicable.

admission reasons and appeal rights. In contrast, Kenya requires family notification of admission reasons but not appeal rights. Egypt's legislation uniquely guarantees patients the right to non-legal advocacy services and is the only Act that explicitly prohibits involuntarily admitted patients from participating in research. Cabo Verde's mental health law allows patients to "accept or refuse to participate in research, clinical trials or training activities." None of the Mental Health Acts makes provisions for Advanced Care Directives (ACDs; legal documents allowing individuals to specify their treatment preferences in advance). However, Nigeria, Ghana, Kenya and Egypt include provisions for supported decision-making, which assists people in making their own decisions rather than substituted decision-making. Nigeria, Cabo Verde, Ghana and Egypt establish judicial or quasi-judicial bodies to assess involuntary admissions and hear appeals. Ghana's legislation uniquely includes a person with lived experience of mental health conditions as a member of the review committee, alongside legal practitioners and healthcare practitioners.

### Regulation of coercive measures in the mental health laws

Table 4 details how coercive measures are regulated across the five countries. All countries, except Cabo Verde, provide specific criteria for implementing coercive measures. All countries permit patient seclusion, though Nigeria, Kenya and Cabo Verde lack clear definitions and descriptions of seclusion procedures. Nigeria has the most detailed regulations for coercive measures, requiring 48 h of prior medical care, written certification from two medical officers, facility accreditation and limiting duration to 72 h or medical necessity. Both Nigeria and Ghana explicitly prohibit using restraint and seclusion as punishment or for staff convenience. Ghana permits ECT only with informed consent or tribunal approval when patients cannot consent. Egypt requires patient education about treatments such as ECT and notably uses the term "Brain Synchronisation Therapy" instead of "Electroconvulsive Therapy" to address stigma (Okasha and Okasha, 2014). Psycho-surgery is explicitly prohibited for involuntary patients in Nigeria

**Table 4.** Regulation of coercive measures in the mental health laws of five African countries

| Coercive measures | Nigeria | Ghana | Kenya | Cabo Verde | Egypt |
|---|---|---|---|---|---|
| Seclusion permitted | Yes | Yes | Yes | Yes | Yes |
| Physical restraint permitted | Yes | Yes | Yes | No | Yes |
| Mechanical restraint permitted | Yes | Yes | Not specified | No | Yes |
| Chemical restraint permitted | Yes | Not mentioned | No | No | No |
| Criteria for coercive measures | After 48 h medical care; certification by two medical officers that restraint/seclusion is the only means to prevent immediate harm; facility accreditation required; cannot exceed 72 h or medical necessity | Imminent danger present; tranquilisation not appropriate or available; facility head or senior nurse authorisation required; immediate documentation | Physical restraint or seclusion only when it is the only means to prevent immediate harm to the person or others | Not mentioned | Dangerous or aggressive behaviour threatening the patient or others; when less restrictive means are insufficient |
| ECT | Yes | Yes (with informed consent or tribunal approval) | Not mentioned | Not mentioned | Yes (termed "Brain Synchronisation Therapy" to reduce stigma) |
| Psychosurgery | Prohibited for involuntary patients | Prohibited for involuntary patients | Not mentioned | Permitted with written consent and two psychiatrist approvals | Not mentioned |

and Ghana, while Cabo Verde permits psychosurgery with appropriate safeguards.

## Policies and regulatory mechanisms for involuntary mental health care

Table 5 outlines the broader policy frameworks and oversight mechanisms governing involuntary mental health care. All countries,

except Cabo Verde, have established regulatory and oversight bodies with national jurisdiction to protect the rights of people with mental health and psychosocial conditions. Only Ghana's mental health law explicitly addresses traditional and alternative medicine, mandating collaboration with the Traditional and Alternative Medicine Council and other providers of culturally relevant mental health care to ensure patient welfare. This includes setting standards for environmental hygiene in mental health facilities, including spiritual centres where traditional healing practices may occur. Egypt maintains both

**Table 5.** Policies and regulatory mechanisms for involuntary mental health care in five African countries

| Regulatory mechanisms | Nigeria | Ghana | Kenya | Cabo Verde | Egypt |
|---|---|---|---|---|---|
| Regulatory and oversight body | Mental Health Assessment Committee (National jurisdiction) | Mental Health Authority (National jurisdiction) | Kenya Board of Mental Health (National jurisdiction) | No | National Council for Mental Health (National). Regional Council for Mental Health (Local jurisdiction) |
| Laws/policies concerning complementary and alternative medicine | No | Yes | No | No | No |
| Regular facility inspections required | Yes | No | Yes | No | Yes |
| Guidance on law interpretation | No | No | No | No | Yes |
| Guidance on minimising compulsory treatments | Yes | No | No | No | Yes |
| Statistics on involuntary treatment ise required | Yes | No | No | No | Yes |

*Note:* CAM: complementary and alternative medicine which in this context includes traditional and faith-based healing practices.

national and local-level oversight bodies. Nigeria, Kenya and Egypt explicitly require regular inspection of mental health facilities, while Ghana and Cabo Verde do not specify this requirement. Only Egypt provides formal guidance on interpreting mental health legislation. Egypt and Nigeria are the only countries with provisions for guidelines on minimising compulsory treatments. Nigeria mandates that facilities implement Ministry-developed guidelines for de-escalating potential crises and managing triggers while respecting patient dignity and human rights. Both Nigeria and Egypt require maintaining statistical records on involuntary treatment use, with Nigeria requiring facilities to maintain registers documenting all instances of seclusion and restraint.

## Discussion

### Alignment with international human rights standards

The analysis of mental health legislation across the five African countries reveals varying degrees of alignment with international human rights standards, particularly the UNCRPD principles. Since ratifying the UNCRPD, Cabo Verde, Egypt, Ghana and Nigeria have repealed outdated colonial mental health laws and enacted new Mental Health Acts that incorporate key human rights principles, such as least restrictive alternatives, last resort measures and the right to appeal. Kenya presents a contrasting trajectory. Until 2022, it operated under the Mental Health Act of 1989, which was itself based on the English Mental Health Act of 1959. The 2022 amendments to Kenya's Act represent minimal modifications to this colonial-era framework rather than comprehensive reform, despite ratifying the UNCRPD in 2008 and issuing policy declarations supporting human rights and community-based care (Di Pierdomenico et al., 2024).

None of the mental health laws were integrated with other laws permitting restrictions on those lacking decision-making capacity ("fusion law"), in order to avoid differential treatment based on disability, as recommended by the United Nations and WHO in their joint 2023 guideline for how to align legislation with international human rights (World Health Organisation and the United Nations [WHO and UN], 2023). Although autonomy-promoting measures, such as ACDs and the right to non-legal advocacy, are largely absent from legislation, some services may provide these supports informally, though this varies significantly by facility and region and lacks legal protection. Supported decision-making is promoted in the mental health laws of all the countries in this survey, except Cabo Verde. The commitment to reducing coercion in mental health care across these countries demonstrates a regional shift towards rights-based approaches, consistent with findings from the previous comparative study in the South Asian context (Dey et al., 2019).

The Concluding Observations for Ghana and Kenya issued by the Committee on the Rights of Persons with Disabilities identified persistent concerns, such as laws permitting involuntary commitment and forced treatment, substitute decision-making regimes, inadequate facility monitoring and insufficient resources (Concluding Observations in Relation to the Initial Report of Kenya | Reword, 2015; CRPD/C/GHA/CO/1: Concluding Observations on the Initial Report of the Ghana | OHCHR, 2024). None of the countries fully complies with the Committee's interpretation that all involuntary commitment constitutes discrimination. Kenya's minimal 2022 amendments show the weakest responsiveness to the 2015 call for a comprehensive law review. This non-compliance demonstrates broader tensions between the Committee's absolutist

standards and implementation realities in resource-constrained, culturally diverse contexts, where community-based voluntary alternatives remain aspirational. The varying pace and extent of these reforms indicate the complex interplay between international standards and local contextual factors.

### Similarities and differences in legislative frameworks

All five countries require family notification for involuntary admissions, reflecting African cultural norms that prioritise collective decision-making and familial involvement in healthcare. This requirement acknowledges the reality that in many African societies, caring for family members with chronic mental health conditions is viewed as a shared responsibility extending to the broader extended family network (Ndlovu and Mokwena, 2023). With limited structured social services across the continent, families bear the primary burden of care responsibilities (Delpy, 2025), giving them significant roles in treatment decisions that may not always align with patient preferences. This creates tension with the UNCRPD's emphasis on individual autonomy, potentially overshadowing patients' voices while raising concerns about those unwilling or unable to rely on family support. The provisions for involuntary treatment, while emphasising family involvement in these countries, may reflect an ethical commitment to communal responsibility for vulnerable members rather than purely paternalistic control. For example, the Yoruba concept of *Omo-olu-iwabi* (a person of dignity and good character) explains how African ethical frameworks conceptualise the relationship between individual autonomy, community responsibility and mental health care. Within this framework, personhood is understood relationally, "a person is a person through other persons," creating reciprocal obligations between individuals and their communities (Ewuoso, 2018).

None of the five countries surveyed has provisions for ACDs in their mental health acts, showing a regional trend. The UNCRPD, through its articles 3 and 25, gives people with mental health conditions the right to make decisions about their own treatment, including the use of ACDs, thus promoting autonomy and respect for personal choice (Convention on the Rights of Persons with Disabilities | OHCHR, 2006). Looking at the broader healthcare setting, advanced healthcare directives are also limited in other healthcare fields across these countries. A lack of awareness and understanding of ACDs among the general population limits their acceptance (Ekore and Lanre-Abass, 2016). While formal ACDs are absent, it is important to recognise that informal advance planning and preference discussions may occur in clinical practice, though these lack the legal protections and enforceability that formal directives provide. Cultural factors significantly influence this situation, as ACDs might be perceived as too individualistic in communitarian societies like those in Africa (Ekore and Lanre-Abass, 2016). In these cultures, collective decision-making processes naturally incorporate multiple perspectives, with individual preferences considered within the broader family and community context, and community or family input being pivotal in health-related decisions. This is underscored by the acceptance of supported decision-making in the Mental Health Acts of all the countries in this survey, except Cabo Verde. This approach recognises that decision-making can be enhanced through family and community support while still preserving the individual's ultimate authority over their care choices. Cultural beliefs that view planning for adverse events as inviting misfortune and stigma surrounding mental health may also

deter individuals from expressing their care preferences for the future.

The study findings indicate that only Ghana mentions CAM in its Mental Health Act, despite CAM's widespread use in African mental health care (James et al., 2018). This gap highlights a significant neglect in integrating traditional healthcare practices into conventional mental health systems. Many African countries, despite moving away from colonial-era laws, still maintain mental health laws and policies that fail to reflect their socio-cultural realities (Akanni and Edozien, 2024; Juma and Ngwena, 2024). A substantial proportion of the population in Africa, particularly in rural areas rely heavily on CAM, such as traditional medicine and spiritual healing. These practices are often the first line of care for persons with mental health conditions due to their accessibility, and alignment with religious and cultural beliefs (Anjorin and Hassan Wada, 2022; Assad et al., 2015; Okasha et al., 2024). By not formally acknowledging the role of traditional healers in mental health care, these governments miss the opportunity to develop culturally sensitive health policies that could improve access to care and protect the rights of people seeking help from CAM practitioners. Globally, there is a growing recognition of CAM in mental health-care. In countries like China and India, traditional medicine is well integrated into mainstream healthcare and mental health services (Thirthalli et al., 2016). Ghana's Mental Health Act mandates collaboration between its mental health authority and the Traditional and Alternative Medicine Council. This formal recognition ensures that CAM practices align with protecting patients' rights and autonomy.

### Contextual factors influencing implementation of the mental health Laws

The emphasis on individual rights, central to international frameworks like the UNCRPD, creates tensions in communitarian African societies where collective decision-making is traditionally valued. Furthermore, the implementation of individual rights-based mental health legislation faces significant challenges due to resource limitations across all five countries. The Committee's Concluding Observations explicitly recognise these challenges, noting Ghana's insufficient allocation of financial and human resources and Kenya's lack of financial coverage for mental health services, while maintaining that resource constraints cannot justify rights violations. The intersection of poverty and healthcare financing creates acute access problems in many African countries, where out-of-pocket payment systems render mental health services inaccessible to most citizens. This problem is pronounced in countries like Nigeria and Kenya, where patients must pay out-of-pocket for mental health services, which are often prohibitively expensive. According to the 2017 Purchasing Power Parity (PPP) data, over 25% of the population in these countries, excluding Cabo Verde, live in extreme poverty. This economic reality creates a cascade of access barriers. Most individuals cannot afford voluntary mental health services, effectively excluding them from preventive or early intervention care. In addition to these economic barriers, many individuals first seek care from traditional and faith healers, who offer more affordable and culturally acceptable alternatives to formal psychiatric services. Treatment in formal psychiatric services typically occurs only when financially capable family members intervene, often at crisis points when the patient's condition has deteriorated significantly and traditional healing approaches have been exhausted (Aluh et al., 2022; Williams et al., 2025). This dynamic perpetuates a system where involuntary admission becomes the predominant pathway to mental health care but paradoxically remains available only to those from families with sufficient economic resources (Aluh et al., 2022). Mental health services in these countries are already overwhelmed and significantly underfunded. The shortage of mental health professionals creates additional implementation challenges. Nigeria, with over 220 million people, has fewer than 300 psychiatrists, while Ghana, Kenya and Cabo Verde face similar shortages. These resource constraints make it difficult to provide community-based, voluntary mental health services that might reduce reliance on coercive measures, creating a gap between legislative aspirations and practical implementation.

Beyond resource limitations, the implementation challenges may also reflect deeper philosophical tensions between the individualistic assumptions embedded in international human rights frameworks and communitarian values observed across various African contexts (Imafidon, 2021). The absolutist interpretation of legal capacity by the General Committee and its prohibition of all involuntary treatment assumes a Western liberal conception of personhood as fundamentally autonomous and independent conflicting with African philosophical traditions where individuals understood as intrinsically connected to their communities and bearing reciprocal care obligations (Ikuenobe, 2015; Ewuoso, 2018; Jansen, 2025). These different philosophical foundations create practical dilemmas for lawmakers attempting to comply with international standards while remaining responsive to local cultural values that emphasise collective responsibility for vulnerable community members.

While all five countries permit restrictive measures and coercive practices in their mental health legislation, the regulatory frameworks governing these practices vary significantly in clarity and practicality. Ghana's law allowing involuntary seclusion only when tranquilisation is not feasible creates conceptual confusion about what constitutes coercive treatment. Nigeria's requirement for 48 h of medical care before implementing coercive measures raises practical questions about how care can be provided to unwilling patients without some form of initial coercion (Akanni and Edozien, 2024).

The relationship between broader human rights contexts and mental health legislation reveals complex patterns that challenge simple assumptions about democratic governance and rights protection. Egypt, despite having the lowest human rights index score among the five countries, demonstrates some of the most robust mental health policies and regulatory mechanisms. Conversely, Cabo Verde, with the highest human rights index score, lacks laws establishing regulatory and oversight bodies for mental health care. This paradox suggests that mental health law reform may be influenced by factors beyond general democratic governance like specific advocacy efforts, professional leadership and targeted policy initiatives. The disconnect between overall human rights contexts and mental health-specific protections underscores the need for focused attention on mental health rights as a distinct area of legislative development. This finding suggests that mental health legislative reform may be driven by factors distinct from general human rights protections, warranting further investigation into the specific mechanisms, such as professional advocacy, international technical assistance and targeted policy initiatives, that shape mental health law development.

Ghana's mental health legislation stands out for its innovative inclusion of service users in mental health review tribunals, representing a significant advancement in promoting civic participation of people with mental health conditions. This approach directly

aligns with the UNCRPD's emphasis on inclusivity and participation, providing people with mental health conditions a meaningful voice in decision-making processes that affect their care and rights. Despite allocating <2% of its health budget to mental health, Egypt has established systematic data collection and reporting requirements that promote transparency and accountability. This approach enables data-driven policy adjustments and demonstrates that effective monitoring is achievable even under resource constraints when supported by political will and appropriate legislation. Moreover, the high prevalence of coercive practices in high-income countries with more mental health resources indicates that merely having resources does not ensure high-quality, human rights-based care, underscoring the need for targeted efforts to protect patients' rights and reduce coercion (Sashidharan et al., 2019).

## Strengths and limitations

This study is among the few that examine the mental health laws across various African countries, considering the unique contextual factors present in each. To ensure consistency in the data collected, a standardised tool was employed across all countries, while multiple credible sources were used to gather additional contextual information. However, a significant limitation of this study is its focus on written law, which often differs from actual practices. While the study benefits from geographical representation across different African regions, it is important to acknowledge that our sampling approach had limitations., First, the study does not include representation from Francophone African countries, which constitute a significant portion of the continent and possess distinct colonial legal traditions shaped by French civil law. This omission limits the generalisability of findings across Africa's diverse legal landscapes. Second, the research team did not include people with lived experience of mental health conditions, whose perspectives would have enriched the analysis by providing insights into the practical implementation and human rights implications of mental health legislation from a service user standpoint. Third, the reliance on professional networks may have inadvertently favoured countries with more established mental health advocacy communities or research infrastructure, potentially excluding countries with less visible but equally important legislative developments. Additionally, the rapidly evolving nature of mental health law reform means that legislative changes may have occurred since the completion of this study. Future research should incorporate implementation studies, stakeholder perspectives – including service users – and longitudinal analyses to better understand the practical impact of legislative reforms on the lives of people with mental health conditions across Africa.

## Conclusion

The study findings show that while significant progress has been made in aligning African mental health legislation with international standards, substantial challenges remain in bridging the gap between written law and practical implementation. The UNCRPD offers a comprehensive framework for revising mental health legislation with an emphasis on consent and autonomy. However, it does not offer specific guidelines for implementing these laws, particularly in resource-limited settings and non-Western communitarian cultures (Aluh et al., 2022).

Mental health law development in Africa must navigate the tension between universal human rights principles and local cultural values while addressing practical constraints related to poverty, resource limitations and service delivery capacity. The innovative approaches demonstrated by individual countries, such as Ghana's inclusion of service users in tribunals and Egypt's monitoring systems, provide models that could be adapted across the region. The integration of traditional healing practices into formal mental health systems represents an important opportunity for developing culturally responsive, rights-based care that acknowledges African therapeutic traditions while maintaining appropriate protections for service users.

**Open peer review.** To view the open peer review materials for this article, please visit http://doi.org/10.1017/gmh.2026.10141.

**Data availability statement.** All data can be provided by the corresponding author.

**Acknowledgements.** The authors would like to gratefully acknowledge Professor Jorun Rugkåsa for her valuable feedback on the initial draft of this manuscript.

**Author contribution.** DOA conceived the study, analysed the data and wrote the initial draft of the manuscript. WJI contributed to writing the first draft of the manuscript and to analysing the data. ENQ, AM, BO and AAS contributed to data collection and to the manuscript draft. All authors reviewed and approved the final draft of the manuscript.

**Competing interests.** The authors declare none.

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
