## [Reviewer Report]

General assessment

The manuscript presents a timely and valuable comparative analysis of mental health legislation in five African countries, which enabled analysis across diverse legal traditions and colonial influences that continue to shape contemporary mental health legislation. The manuscript adopts a rights-based lens aligned with the CRPD and highlights legal progress, ongoing gaps, and the notable absence of Advance Care Directives (ACDs) across the jurisdictions studied. The paper offers useful insights for both academic and policy audiences and fills a clear gap in the literature on African mental health law.

Strengths

Comparative structure: The country-by-country format is effective for organizing key differences and similarities in legislation.

Empirical contribution: The inclusion of an empirical study adds practical value to the legal analysis. It enriches the manuscript by shedding light on how mental health laws operate in practice and reveals implementation challenges that may not be apparent from legal texts alone. This is a significant strength, given the scarcity of empirical work in African legal scholarship.

Rights-based framework: The manuscript is grounded in international human rights norms, particularly the CRPD, which strengthens its normative foundation.

Clarity of argument: The author clearly identifies the absence of ACDs and contextualizes this within broader legal and institutional limitations.

Engagement with cultural contexts: The discussion of cultural values is handled with nuance, acknowledging their influence without essentialising African legal systems or traditions.

Specific comments and suggestions for improvement

African Disability Protocol and CRPD – entry into force and ratification Status: Please indicate the date on which the African Disability Protocol officially entered into force. It is also recommended that this information, along with the ratification status of both the ADP and the UNCRPD for each of the five comparative countries, be incorporated into the “Study Design and Country Selection” subsection. This will strengthen the legal and institutional context of the analysis and help clarify the extent of each country’s formal commitment to disability rights instruments.

Kindly also ensure that accurate terminology is used throughout—refer to the ‘adoption’, ‘entry into force’, or ‘ratification’ of international instruments rather than ‘enactment’, which is not typically used in relation to treaties.

Institutional Accuracy – Kenya Mental Health Act: The manuscript refers to the Kenya National Commission on Human Rights (KNCHR) in the context of the Mental Health Act. However, the Act does not explicitly mention KNCHR. Instead, the Kenya Board of Mental Health is the statutory body established under the Act to oversee mental health matters. Kindly revise this section to accurately reflect the institutional framework provided for in the legislation.

Terminology consistency: The manuscript generally uses respectful, person-centered language. However, consistency is important—terms like ‘people with mental health and psychosocial disabilities’ should be used uniformly, avoiding acronyms such as ‘PMHCs’ unless directly quoted.

Alignment Between results and discussions: Please ensure that the discussions presented in the body of the manuscript is clearly aligned with the results. Any claims or interpretations in the results section should be directly supported by evidence and arguments developed in the discussions. For example, under the discussion on ‘Similarities and Differences in Legislative Frameworks’ the authors have indicated that ‘All five countries require family notification for involuntary admissions’, which is not the case as per the results obtained. Strengthening this alignment will improve the internal coherence and persuasiveness of the paper.

Referencing a few unsupported assertions: While the manuscript is generally well-referenced, there are a few assertions that would benefit from source attribution. For example, the claim that ‘[t]reatment typically occurs only when financially capable family members intervene, often at crisis points when the patient’s condition has deteriorated significantly’ under the ‘Contextual Factors Influencing Implementation of the mental health laws’ sub-section. Kindly review and cite appropriate references for these specific claims to enhance academic rigour and credibility.

Minor notes

Spell out acronyms at first mention, even in tables or figures.

Use ‘Cabo Verde’ consistently, as it is the official name recognized by the UN.

There appears to be an inconsistency in the keywords provided—two different sets of keywords are listed in different parts of the manuscript. Kindly ensure that the keywords are consistent and accurately reflect the central themes of the paper.

Conclusion

This manuscript makes a valuable contribution to African human rights and mental health law literature. Its empirical grounding, comparative framing, and rights-based approach make it well-suited for publication, pending minor revisions. With modest clarifications, the article will serve as a strong academic and policy resource.

Specific comments

Title - Consider using ‘Cabo’ instead of ‘Cape’. The former is the official name. Change throughout the paper.

Keywords - align with content

pg 3 ln28 - “rights of those”, change to persons

pg3 ln 29 - CRPD - UNCRPD?

pg 3 ln 38 - Peoples'

pg3 ln 40 - “While the Charter does not explicitly address mental health” - This is not accurate. May be say ‘it does not provide detailed guidance on mental health’

Pg 3 ln 43 - Indicate the date when it entered into force and whether it has been ratified by the five countries.

pg 3 ln 50 - “ As most African countries” - be more specific

pg 3 ln 51 - “its” - what?

pg 4 ln 25 - define GDP

pg 4 ln25 - capital S for South Asia

pg 5, table 1 MHS and MHC - First mention. Spell out acronyms

Pg 6, ln 18 - define WHO ATLAS

Pg 6, ln 36 - Avoid use of acronyms when referring to persons with disabilities

pg 9 ln5 - capital A for Act

pg 10 ln 32 - ECT - First mention. Spell it out.

table 5 - Kenya Board of Mental Health? KNCHR is not provided for under the Mental Health Act of Kenya.

Pg 12, ln 19 - Enactment - Not accurate. Substitute with adoption or ratification.

pg 12, ln 39 - “ All 5 countries” - Except Nigeria. See table 3.

pg 12, ln 50 - ACDs - what are these?

pg 13, ln 15 - capitalise Mental Health Act

pg 13, ln 44-45 - provide source

pg 14, ln 8, mental health act - Capitalize first letters since you are referring to a law of a particular country

---

## [Reviewer Report]

1. General overview

This research provides insight into five examples of mental health legislation in African countries. However, as the authors attempt to cover a wide range of findings, the analysis tends to be somewhat superficial. I was particularly interested in reading about pivotal and unique legal provisions within the legislation. Due to the limited depth of the findings, it is difficult to identify which provisions from each country’s mental health legislation align with the World Health Organization’s Mental Health Guidance and other authoritative documents. To enhance the manuscript, I offer the following recommendations.

2. Constructive feedback

a. Justification for country selection

I am curious about the rationale behind selecting Nigeria, Egypt, Ghana, Cape Verde, and Kenya for this study. It would be helpful if the author could explain the criteria or reasoning for choosing these five countries.

b. Limited use of literature

The manuscript would benefit from engaging more extensively with international human rights law beyond the Convention on the Rights of Persons with Disabilities. The Committee on the Rights of Persons with Disabilities has issued various soft law instruments, such as General Comments and Concluding Observations. I recommend that the author explore and incorporate these documents to strengthen the legal analysis.

c. Adaptation of the FOSTREN* instrument

Using the FOSTREN* instrument as a methodological framework is a clever approach, as it contributes additional data on mental health legislation. However, the author should clarify what specific adaptations were made to the instrument and how these differ from its original use in the European context. This explanation would help readers understand the uniqueness and relevance of the adapted version.

d. Inclusion of Concluding Observations and other authoritative documents from Human Rights Treaty Bodies

I suggest that the author refer to the Concluding Observations issued for the five African countries studied. These documents could provide valuable perspectives, particularly regarding whether the recommendations from Human Rights Treaty Bodies have been implemented. This addition would enrich the section titled “Contextual Factors Influencing Implementation of the Mental Health Laws” (pp. 13-14).

---

## [Reviewer Report]

The paper is a valuable contribution to the global discourse on mental healthcare and human rights. Of special importance are the following findings.

1.The comparative analysis of mental health laws in five African countries reveals complex interactions and tensions between international human rights frameworks and local realities that are crucial at the level of implermentaltion of mental health policies.

2. Another important finding is about complex patterns that challenge simple assumptions about level of democratci governence and protection of human rigthts in mental healthcare. The paradox is highlighted that more comprehensive mental health policies may be present in the countries with lower mental health index, while countries with higher mental health index may be lagging behind with mental health laws and policies. Further analysis of this paradox may be very useful for the global discourse.

3. Cultural issues such as reliance on collective decision making cultures, and their analysis is another valuable contribution to the global discourse. This is important with regard to tensions while aligning national contexts to international human rights standards and principles (such as individual autonomy and empowerment), enshrined in the CRPD convention,

The aforementioned and other findings based on the good quality research, with subsequent discussion and conclusions, provide an obvious added value to the very much needed discourse on present and future of global mental health.

---

## [Reviewer Report]

This manuscript provides a valuable comparative analysis of mental health laws across five African countries. The topic is important and the study demonstrates a significant effort in compiling legal and policy information. Overall, the manuscript is informative, but several areas still require clarification, improved consistency, and smoother flow to strengthen the argument and readability.

Specific Comments

1. Impact Statement: The manuscript appears to imply that the CRPD is a Western-developed instrument. This is not accurate and should be clarified.

2. Abstract: Consider using ‘CRPD standards’ instead of ‘international human rights standards’ for precision and consistency.

3. Introduction: The statement suggesting that the CRPD is aligned with the African Disability Protocol contradicts the claim that it (CRPD) is a Western-developed instrument. Ensure consistency with points made in the impact statement section, on page 3, and on page 14.

4. Page 3: Clarify the term ‘older laws’. Ghana’s mental health law was enacted in 2012, and there is no indication that it has been changed since.

5. Methods – Countries’ Context: Align the text with the table. The manuscript states that Kenya has the highest poverty rate (36.1%), but the table shows Egypt at 80%. Please verify and correct any discrepancies.

6. Regulation of Involuntary Admission: Capitalize the ‘A’ in ‘act’ consistently, including on page 9.

7. Policies and Regulatory Mechanisms for Involuntary Mental Health Care: The phrase ‘including’ is repetitive and awkward. Consider rewording to avoid redundancy.

8. Alignment with International Human Rights Standards: Colonial-era legislation referred to on page 12 should be accurately reflected in Table 2. Currently, the table refers to ‘The Mental Health (Amendment) Act, 2022’.

9. Similarities and Differences in Legislative Frameworks: Page 13: Consider using ‘relationally’ instead of ‘relatively’. Page 13.

The discussion of ‘ACDs’ has already been addressed under the sub-section “Alignment with International Human Rights Standards” and may be relocated there for clarity. Page 13 paragraph 2.

10. Contextual Factors Influencing Implementation of Mental Health Laws:

Page 14, paragraph 2: The first part fits better under the sub-section ‘Similarities and Differences in Legislative Frameworks’.

Terms such as ‘African communitarian ethics’ and ‘African philosophical traditions’ suggest a unified, homogenous category, which is misleading. Consider instead referring to recurring patterns or tendencies across different African contexts.

The second part of the paragraph, beginning with ‘While all five’, appears disconnected from the first part and requires better cohesion.

11. Strengths and Limitations: The sentence beginning with ‘Although the study aimed for’ reads like the start of a new subsection and disrupts flow. Consider integrating it more smoothly with the preceding text.

12. Clarity: Rephrase some sentences for clarity and check the tense. For instance in page 12.

13. Comments to the author(s): Please refer to the attached file for detailed, line-by-line comments and suggestions for improvement.

---

## [Reviewer Report]

The author has revised the manuscript according to my feedback. Additional legal materials have been incorporated, and the overall academic quality of the legal materials has improved. Thank you.